# Silencing of circCRIM1 Drives IGF2BP1-Mediated NSCLC Immune Evasion

**DOI:** 10.3390/cells12020273

**Published:** 2023-01-10

**Authors:** Wenbei Peng, Linlin Ye, Qianqian Xue, Xiaoshan Wei, Zihao Wang, Xuan Xiang, Siyu Zhang, Pei Zhang, Haolei Wang, Qiong Zhou

**Affiliations:** Department of Respiratory and Critical Care Medicine, Union Hospital, Tongji Medical College, Huazhong University of Science and Technology, Wuhan 430022, China

**Keywords:** circular RNAs, immune evasion, NSCLC

## Abstract

Objectives: Circular RNAs (circRNAs) have been found to have significant impacts on non-small cell lung cancer (NSCLC) progression through various mechanisms. However, the mechanism of circRNAs modulating tumor immune evasion in NSCLC has yet to be well-revealed. Materials and Methods: Through analyzing the expression profiles of circRNAs in NSCLC tissues, RNA FISH, pull-down assay, mass spectrometry analysis, and RIP, circCRIM1 was identified, and its interaction with IGF2BP1 was confirmed. The effects of circCRIM1 on modulating tumor immune evasion were explored via co-culture in vitro and in tumor xenograft models. Subsequently, we evaluated the regulatory effects of circCRIM1 on IGF2BP1 and screened its target genes through RNA sequencing. Finally, we explored the underlying molecular mechanisms that circCRIM1 could regulate the stability of target mRNA. Results: circCRIM1 was downregulated in NSCLC, and its expression was positively correlated with favorable prognoses. Furthermore, circCRIM1 was more stable than its linear transcript and was mainly localized in the cytoplasm. Mechanistically, circCRIM1 destabilized HLA-F mRNA via competitive binding to IGF2BP1. Importantly, the overexpression of circCRIM1 suppressed the immune evasion of NSCLC and promoted the expressions of Granzyme B, IFN-γ, and TNF-α of CD8+ T and NK cell in vitro co-culture assays and tumor xenograft models. Conclusions: This study identifies circCRIM1 as a new tumor suppressor that inhibits tumor immune evasion through a competitive combination with IGF2BP1 to destabilize HLA-F mRNA.

## 1. Introduction

Lung cancer is one of the most commonly diagnosed cancers, and it is the leading cause of cancer-related death worldwide [1]. Non-small cell lung cancer (NSCLC) is the most common type, including lung adenocarcinomas (LUADs), lung squamous cell carcinomas (LUSCs), and others. In recent years, significant progress has been made in the treatment of NSCLC, especially in immune checkpoint therapies, such as the antibody-based blockade of programmed death 1 and programmed death-ligand 1 (PD-[L]1) [2,3]. Nevertheless, only a minority of patients achieved durable benefits from PD-[L]1 blockade [4]. Therefore, it is necessary to explore the underlying interaction mechanism between lung cancer cells and immune cells in order to discover new targets for immunotherapy.

Circular RNAs (circRNAs) are a newly discovered class of non-coding RNAs (ncRNAs). They are generated from the back-splicing of precursor mRNA to form covalently closed transcripts [5,6,7,8,9]. Previous studies revealed the significant roles of circRNAs in various cancers, such as in promoting the proliferation and metastases of tumors, activating vascular regeneration, and tumor immune evasion [10,11,12,13]. CircRNAs exert their functions by serving as miRNA sponges, interacting with RNA-binding proteins, or being translated into short peptides. However, most research on circRNAs reported in lung cancer has mainly focused on the role of tumor proliferation, invasion, and metastasis. It was only in recent years that circRNAs were found to play a potential role in regulating tumor immunity. For instance, in lung cancer, circRNA-002178 could enhance PDL1 expression to induce T-cell exhaustion via sponging miR-34 [14]. circUHRF1 upregulated the expression of TIM-3 via sponging of miR-449c-5p to inhibit NK cell function in hepatocellular carcinoma [15]. Nevertheless, the mechanism of circRNAs in modulating the interactions between tumors and immune cells in lung cancer still requires further revelation.

The insulin-like growth factor-2 mRNA-binding protein 1 (IGF2BP1) is a member of the conserved IGF2BP proteins family, which possesses similar biochemical functions for the resemblance of structures [16]. Recent studies indicate that IGF2BP1 could facilitate target gene expression by enhancing the stability and translation of mRNA in an N6-methyladenosine (m6A)-dependent manner [17]. Additionally, circRNAs have been demonstrated to participate in regulating the expression and function of IGF2BP1 [18,19,20,21,22]. For example, circPTPRA interacted with IGF2BP1 to suppress bladder cancer progression by blocking the recognition of RNA N6-methyladenosine [22]. circNDUFB2 inhibited non-small cell lung cancer progression and activated antitumor immunity by forming the TRIM25/circNDUFB2/IGF2BPs ternary complex [18]. However, it remains poorly understood how circRNAs interact with IGF2BP1 in regulating tumor immune evasion of NSCLC.

Human leukocyte antigen F (HLA-F) belongs to the nonclassical HLA class I molecule and possesses a similar structure to HLA-Ia molecules (HLA-A, HLA-B, and HLA-C) [23,24]. HLA-F has been found to mainly play a role in immune tolerance and tumor progression that is distinct from the role of presenting antigens of HLA-Ia molecules. Notably, high expression of HLA-F was related to the poor overall survival of patients with lung cancer, glioma, hepatocellular carcinoma, and nasopharyngeal carcinoma [25,26,27,28]. HLA-F may activate or inhibit NK and CD8+T cells by physically interacting with killer-cell immunoglobulin-like receptors (KIRs) such as KIR3DL2, KIR2DS4, and KIR3DS1, indicating that HLA-F may potentially contribute to tumor immune evasion [29,30]. Nevertheless, the role and mechanism of HLA-F in the immune evasion of NSCLC still remain to be clarified.

In this study, we discovered that circCRIM1, which is generated from the cyclization of the CRIM1 gene, was significantly downregulated in NSCLC. Subsequent studies showed that circCRIM1 could interact with IGF2BP1 to negatively modulate the immune evasion of NSCLC by destabilizing HLA-F mRNA. Our findings provided a novel mechanism of circCRIM1/IGF2BP1-mediated regulation of immune evasion and indicated that circCRIM1 and IGF2BP1 may act as promising immune therapeutic targets for NSCLC.

## 2. Materials and Methods

### 2.1. Tissue Specimen Collection

Ninety paired primary NSCLC tumorous tissues and adjacent normal tissues of patients were obtained from surgical resection at the Department of thoracic surgery of the Union Hospital of Tongji Medical College (Wuhan, China) between January 2016 and March 2021. Histological and pathological reports of the specimens were diagnosed by at least two professional clinical pathologists. After surgical removal, all tissue specimens were immediately snap-frozen in liquid nitrogen and then stored at −80 °C. All specimens were obtained with the consent of patients, and the study was approved by the Institutional Review Board of Tongji Medical College of Huazhong University of Science and Technology. All patients were followed up regularly in order to obtain the overall survival (OS) time.

### 2.2. Cell Culture and Treatment

NSCLC cell lines (A549, H1299, and Calu1), human bronchial epithelial cell line (BEAS-2B), and murine lung carcinoma cell line (LLC) were purchased from the American Type Culture Collection (ATCC, Manassas, VA, USA). A549 cells were grown in the F-12K (Gibco) medium supplemented with 10% FBS (Gibco) and 1% penicillin/streptomycin. H1299 and Calu1 cells were cultured in the RPMI-1640 medium (Gibco) with 10% FBS and 1% penicillin/streptomycin. BEAS-2B and LLC cells were maintained in Dulbecco’s modified Eagle’s medium (DMEM) supplemented with 10% FBS and 1% penicillin/streptomycin. All cells were cultured at 37 °C with 5% CO_2_ in a humidified incubator.

### 2.3. RNA Extraction and qRT-PCR

Total RNA was isolated from cells or tissues using the miRNeasy Mini Kit (Qiagen), and cDNA was synthesized with HiScript II Q RT SuperMix for qPCR (Vazyme, Nanjing, China). The real-time PCR analyses were performed using AceQ qPCR SYBR Green Master Mix (Vazyme) by the StepOnePlus Real-Time PCR System (Applied Biosystems, Foster City, CA), and the gene expressions were normalized with GAPDH. The primers are listed in Appendix A.

### 2.4. RNase R Treatment

A total of 1 μg of RNA was incubated for 15 min at 37 °C, with or without the treatment of 5 U RNase R (Epicentre Technologies, Madison, WI, USA); then, RNA samples were analyzed using qRT-PCR.

### 2.5. Actinomycin D Treatment and RNA Stability Assay

A549 and Calu1 cells were planted into six-well plates and treated with 5 μg/mL Actinomycin D (Sigma) or DMSO after 24 h, and then collected at the indicated time points. The RNA was extracted and analyzed using qRT-PCR. The half-life of RNA was evaluated according to a previous study [17].

### 2.6. Nuclear and Cytoplasmic Extraction

Cytoplasmic and nuclear fractions were isolated according to the manufacturer of the PARIS™ Kit (AM1556, Thermo Fisher Scientific, Waltham, MA, USA). Cells were lysed in a cell fraction buffer on ice for 10 min and then centrifuged at 400× *g* for 5 min at 4 °C. The supernatant was collected as a cytoplasmic fraction, and the nuclei pellets were collected by washing them with cell fraction buffer.

### 2.7. Vector Construction and Cell Transfection

Human cDNAs were synthesized using TSINGKE (Wuhan, China) to construct circCRIM1, IGF2BP1, and HLA-F overexpression plasmids. The circCRIM1 was cloned into the pcDNA3.1(+) CircRNA Mini Vector (addgene #60648); in addition, HLA-F and full-length or truncated proteins of IGF2BP1 (Appendix A) were cloned into the p3XFLAG-CMV-10 vector (Sigma-Aldrich). Specific short hairpin RNAs (shRNAs) for circCRIM1 were cloned into pLKO.1-puro (Sigma-Aldrich). Transfection was performed using Lipofectamine 3000 (Life Technologies) according to the manufacturer’s instructions. Neomycin or puromycin (Invitrogen) was used to screen stably transfected cell lines. An empty vector was applied as a control (Appendix A).

### 2.8. Isolation and Activation of CD8 + T Cells and NK Cells

Human peripheral blood mononuclear cells (Hu-pbmcs) were isolated from the whole blood of healthy volunteers using Lymphoprep (StemCell Technologies, Vancouver, BC, Canada) according to the manufacturer’s instructions. CD8+T or NK cells were separated from freshly extracted Hu-pbmc using magnetic separation kits (130-096-495 or 130-092-657, Miltenyi). After isolation, CD8+T cells were activated and expanded with CD3/CD28 beads (#11131D, ThermoFisher Scientific) for 2 to 3 days in the RPMI 1640 medium containing human rIL-2 (20 U/mL, #PHC0021, ThermoFisher Scientific) and 10% heat-inactivated FBS. Activated CD8+T cells were cultured for 4–6 days before use in all experiments. NK cells were activated by culturing in the RPMI 1640 medium containing rIL-2 (100 U/mL) and rIL-15 (20 ng/mL, #PHC9151, ThermoFisher Scientific).

### 2.9. In Vitro Co-Culture Assay

A549 cells were first labeled with CFSE (5 ng/mL, #C34554, Invitrogen), according to the manufacturer’s instructions, which then were seeded at 40,000 cells per 24 wells. The next day, cells were co-cultured with activated CD8+T or NK cells at the indicated E:T (Effect-to-target) ratios for 24 h. The dead tumor cells were detected via propidium iodide staining (#556463, BD Biosciences), using flow cytometry.

### 2.10. RNA Pull-Down Assays and Mass Spectrometry Analysis

Biotin-labelled control (antisense) and circCRIM1 (sense) probes (Appendix A) were synthesized by TSINGKE. RNA pull-down assays were conducted as described previously [18]. The bound proteins in the pull-down samples were detected using Western blotting or mass spectrometry. Silver staining was performed according to the manufacturer’s instructions of the PAGE Gel Silver Staining Kit (Solarbio, Beijing, China). Mass spectrometry analysis was accomplished by Novogene (Tianjin, China) using an EASY-nLCTM 1200 UHPLC system (Thermo Fisher Scientific) coupled with a Q ExactiveTM series mass spectrometer (Thermo Fisher Scientific). Furthermore, protein identification and quantification were performed by Proteome Discoverer software (version 2.2, ThermoFisher Scientific).

### 2.11. Western Blot and RNA Immunoprecipitation (RIP)

Western blot assays were conducted as described previously [21]. Quantitative analysis was performed using ImageJ. The band intensity for proteins of interest was normalized to the band intensity of controls. RIP assays were performed according to the manufacturer’s instructions of the Magna RIP RNA-Binding Protein Immunoprecipitation Kit (Millipore, Bedford, MA, USA). Antibodies used included antibodies against IGF2BP1 (#8482, Cell Signaling Technologies), IGF2BP2 (#14672, Cell Signaling Technologies), IGF2BP3 (#57145, Cell Signaling Technologies), FLAG (#F3165, sigma), Rabbit IgG (#31235, Thermo Fisher Scientific), HLA-F (#ab126624, Abcam), GAPDH (#60004-1-Ig, ProteinTech Group, Chicago, IL, USA), HRP-conjugated secondary goat anti-mouse (#SA00001-1, Proteintech), and goat anti-rabbit (#SA00001-2, Proteintech).

### 2.12. Fluorescent In Situ Hybridization (FISH) and Immunofluorescence

Cy3-labelled circCRIM1 probes were synthesized by TSINGKE (Appendix A). FISH and immunofluorescence assays were conducted as described previously [21]. 

### 2.13. Dual-Luciferase Reporter Assay

DNA fragments of wild-type and mutant HLA-F mRNA were synthesized by TSINGKE and cloned into the pMIR-REPORT vector (Promega, USA) to construct firefly luciferase reporters. The firefly luciferase reporter plasmids (pMIR-wt and pMIR-mut) and renilla luciferase reporter control vectors (pRL-TK) were co-transfected with circCRIM1 plasmid or vectors in order to detect luciferase activities. Moreover, the pMIR-wt reporter plasmids were co-transfected with circCRIM1 and IGF2BP1 overexpression plasmids using Lipofectamine 3000. The firefly and renilla luciferase activities were measured after 48 h with the Dual-Luciferase Reporter Assay system (Promega, Madison, WI, USA).

### 2.14. In Vivo Tumorigenesis Assays

All animal experiments were approved by the Animal Care Committee of Tongji Medical College. The 6–8-week-old female C57BL/6 and NOD-SCID IL2Rγ-null (NSG) mice were obtained from Beijing Vital River Laboratory Animal Technology Co., Ltd.; they were housed in a specific-pathogen-free facility. A total of 2 × 10^7^ human pbmcs were injected intravenously into NSG mice to establish humanized NSG mouse models. One week after Hu-pbmcs transplantation, the mice were subcutaneously injected in the right flanks with A549 cells that were stably transfected with the vector or circCRIM1. Body weights and tumor sizes of the mice were measured every 4 to 5 days. Tumors were measured with calipers and calculated using the following formula: v = width^2^ × length × 0.5. The mice were then sacrificed, and tumors were removed and weighed at the end of the experiment.

### 2.15. Flow Cytometry Analysis

Fresh tumor tissues were washed with PBS, cut into small pieces, and digested with collagenase type IV (1 mg/mL, Sigma) and DNase I (10 μg/mL, Sigma) in the RPMI 1640 medium for 30 min at 37 °C. Tumor suspensions were passed through a 70 μm filter. Cell suspensions and tumor suspensions were surface-labeled for antibodies for 30 min at 4 °C. For intracellular staining, cells were fixed and permeabilized using a Transcription Factor Buffer Set according to the manufacturer’s instructions (#562574, BD Biosciences) and incubated with antibodies for 30 min at 4 °C. The stained cells were analyzed on an LSRFortessa X-20 (BD Biosciences). The specific antibodies were used for flow cytometry as follows: Anti-Human CD8-FITC (SK1, #344704, Biolegend), CD56-PE/Cy7 (B159, #557747, BD Biosciences), Granzyme B-Brilliant Violet 421 (QA18A28, #396414, Biolegend), IFN-γ-APC (4S.B3, #502512, Biolegend), CD3-PerCP/Cyanine5.5 (UCHT1, #300430, Biolegend), CD45-BV510 (HI30, #563204, BD Biosciences), TNF-α-PE (MAb11, #559321, BD Biosciences), HLA-F-APC (3D11, #373208, Biolegend), and Fixable Viability Stain 780 (#565388, BD Biosciences).

### 2.16. Immunohistochemistry and Terminal Deoxyribonucleotide Transferase-Mediated Nick-End Labeling (TUNEL) Analyses

Tumor tissues were fixed with formalin, embedded with paraffin, and stained with immunohistochemical (IHC), as previously described [21]. IHC analyses were performed with primary antibodies that were specific for CD8 (#99746, CST), CD56 (ab81292, Abcam), and HLA-F (ab126624, Abcam). TUNEL assays were performed using the InSitu Cell Death, Fluorescein detection kit (Roche, Basel, Switzerland). For HLA-F staining, high expression was defined as equal to or more than an IHC score of 2. IHC scores were calculated using the following formula: Staining score = sum of each staining intensity (0–3) multiplied by the percentage of positive cells (0–100%). Cells positive for CD8, CD56, and TUNEL were counted in five microscopic fields, and their averages were calculated. Two pathologists independently evaluated the stained slides.

### 2.17. RNA Sequencing 

RNA sequencing and library preparation of A549 and Calu1 cells were performed by Novogene (Tianjin, China). Differentially expressed genes (DEGs) were determined using differential *p* values of <0.05 and an absolute fold change of >1.5 or < 0.5.

### 2.18. Cell Counting Kit-8 (CCK-8), Colony Formation and CFSE Proliferation Assays

For the CCK8 assay, tumor cells were seeded into 96-well cell plates and cultured for different time periods (0 d, 1 d, 2 d, 3 d, and 4 d). Then, 10 μL of the CCK8 solution (Dojindo, Kumamoto, Japan) was added to each well of the plate to incubate for 2 h at 37 °C. The absorbance at 450 nm was measured using a spectrometer (Thermo Fisher Scientific). For the colony formation assay, 700 cells were seeded in 2 mL complete culture media in six-well plates for 1–2 weeks. Then, colonies were washed, fixed, stained, and counted. For the CFSE proliferation assay, tumor cells were stained with CFSE (Invitrogen) according to the manufacturer’s instructions and cultured. Then, cells were analyzed on an LSRFortessa X-20 (BD Biosciences, Franklin Lakes, NJ, USA).

### 2.19. Transwell Migration and Apoptosis Assays

For the migration assay, cells were plated in upper Transwell chambers (Corning, NY, USA) with serum-free medium, and the medium supplemented with 10% FBS was placed in the lower chambers. After incubation for 24 h, the migrated cells were fixed, stained, and counted. For the apoptosis assay, cells were seeded into a six-well plate. The cell apoptosis assay was performed according to the manual of the FITC Annexin V Apoptosis Detection Kit I (BD Biosciences). Data were analyzed by FlowJo software (FlowJo, Ashland, OR, USA).

### 2.20. Statistical Analysis

Data were expressed as mean ± SD. Statistical analyses were performed as described in the Figure legends using Prism 8.1.2 (GraphPad Software Inc, La Jolla, CA, USA). Statistical significance was determined by the unpaired Student’s *t*-test, paired Student’s *t*-test, one-way ANOVA with Tukey’s post-test, and two-way ANOVA with Tukey’s post-test. Kaplan–Meier survival curves were calculated using a log-rank test. Statistical significance was established at the levels of *, *p* < 0.05; **, *p* < 0.01; and ***, *p* < 0.001.

## 3. Results

### 3.1. Identification of circCRIM1 as a Candidate NSCLC Suppressor

Our previous studies found that hundreds of circular RNAs were differentially expressed in NSCLC via high-throughput sequencing (NCBI accession number: PRJNA863919). Among these circRNAs, we noted that circCRIM1 (hsa_circ_0002346) was significantly downregulated in NSCLC. Furthermore, we used qRT-PCR to verify the expression of circCRIM1 in 90 pairs of NSCLC tissues and matched adjacent normal lung tissues. Consistently, circCRIM1 was significantly downregulated in NSCLC tissues and cell lines, but the expression of CRIM1 mRNA (mCRIM1) showed no significant difference (Figure 1A,B and Appendix A). Moreover, the low abundance of circCRIM1 in NSCLC patients may be associated with poor overall survival (OS), while the expression of mCRIM1 was not related to survival times (Figure 1C and Appendix A). Taken together, these results suggest that circCRIM1 loss may contribute to NSCLC progression, and circCRIM1 could serve as a prognostic biomarker for NSCLC patients.

### 3.2. Characterization of circCRIM1 in NSCLC Cells

By mapping the human reference genome (GRCh37/hg19), we identified that circCRIM1 (hsa_circ_0002346, chr2:36623756-36669878) is derived from exons 2, 3, and 4 of the cysteine-rich transmembrane BMP regulator 1 (CRIM1) gene, which is located on chromosome 2p22.2 based on the known circRNA database (circBase). The divergent and convergent primers were designed to amplify circCRIM1 and mCRIM1, respectively. The cyclization site of circCRIM1 was validated by the amplification of divergent primers and Sanger sequencing (Figure 1D). The results showed that circCRIM1 was only amplified in cDNA from A549 and Calu1 cell lines, while CRIM1 mRNA was detectable in both genomic DNA (gDNA) and cDNA (Figure 1E). GAPDH acted as a control. 

In addition, the RNase R digestion assay and Actinomycin D assay demonstrated that circCRIM1 was more resistant and had a longer half-life in contrast to the CRIM1 mRNA in A549 and Calu1 cells, indicating that circCRIM1 was more stable (Figure 1F,G). According to nuclear and cytoplasmic fraction assays and the RNA fluorescence in situ hybridization (FISH) assay, circCRIM1 was predominantly localized in the cytoplasm (Figure 1H,I and Appendix A). Collectively, these results demonstrated that circCRIM1 was more stable in circular form and was mainly distributed in the cytoplasm.

### 3.3. CircCRIM1 Modulation Alters NSCLC Cells Immune Evasion In Vitro 

To explore the potential biological effect of circCRIM1 on NSCLC cells, we overexpressed circCRIM1 by transfecting it with the circCRIM1 vector and depleted the expression of circCRIM1 using shRNA. The circCRIM1 was successfully overexpressed and silenced in NSCLC cells and was validated with qRT-PCR, while the CRIM1 mRNA was not obviously changed (Appendix A). Functional assays showed that the proliferation, migration, and apoptosis of NSCLC cells were not significantly affected by the overexpression of circCRIM1 (Appendix A). Notably, tumor and cytotoxic immune cell co-culture assays confirmed that overexpression of circCRIM1 increased the proportion of tumor cell death and promoted the expression of Granzyme B, IFN-γ, and TNF-α of CD8+ T and natural killer (NK) cells. In contrast, silencing circCRIM1 decreased the ratio of tumor cell death and weakened the cytotoxic function of CD8+ T cells and NK cells (Figure 2A–H and Appendix A). Together, these data suggested that circCRIM1 could play a potential role in regulating NSCLC cells’ immune evasion and the cytotoxic function of CD8+ T and NK cells.

### 3.4. CircCRIM1–IGF2BP1 Interaction in the Cytoplasm of NSCLC

To further probe the molecular mechanism of circCRIM1 in NSCLC cells, RNA pull-down was performed using biotin-labeled circRNA probes targeted at the junction site (Figure 3A). Mass spectrometry revealed 892 proteins pulled down by both circCRIM1 and the control probe, and only 57 proteins by the circCRIM1 probe. Strikingly, in-depth analysis of mass spectrometry results exhibited candidates interacting with RBPs of circCRIM1, including the IGF2BP family (Figure 3B). Furthermore, the interplay of circCRIM1/IGF2BPs was validated by the precipitates immunoprecipitated and RIP assay (Figure 3C,D and Appendix A). Interestingly, we observed significant enrichment of circCRIM1 in RIP experiments, particularly for IGF2BP1 (Figure 3E and Appendix A). Meanwhile, data from the TCGA database showed that the high expression of IGF2BP1 was associated with inferior survival in patients with NSCLC, but the expressions of IGF2BP2 or IGF2BP3 were not related to overall survival (Appendix A). Moreover, overexpressed IGF2BP1 completely abolished the inhibition of tumor evasion elicited by circCRIM1 overexpression, suggesting that IGF2BP1 may be a critical interacting protein of circCRIM1 (Appendix A). 

To investigate the mechanism of the interactions between circCRIM1 and IGF2BP1, we predicted the possible binding sites using the catRAPID website [31]. Based on the IGF2BP1 functional domains, we performed the protein truncation test (Figure 3F and Appendix A). As far as we know, IGF2BP1 included four K homology (KH) domains and two RNA recognition motifs (RRMs) [32]. While the KH1/2 domains are critical for IGF2BPs to modulate the stability of target mRNAs by binding to the coding region stability determinant (CRD) or cis-determinants in the 3′-UTR [33], KH3/4 domains are necessary for IGF2BP1 to stabilize targeted mRNAs in an m6A-dependent manner [17]. Even though most studies indicated that the KH domains were major in the RNA binding of IGF2BP1, the RRM domains of IGF2BP1 could potentially promote the stability of target mRNAs [34]. In our study, the RNA immunoprecipitation assay further indicated that IGF2BP1 interacted with circCRIM1 by RRM domains, but not other domains (Figure 3G). We confirmed the colocalization of circCRIM1 and IGF2BP1 in the cytoplasm via immunofluorescence in situ hybridization (IF-FISH) assays (Figure 3H). In summary, these results proposed that IGF2BP1 interacted with circCRIM1 by RRM domains in the cytoplasm of NSCLC.

### 3.5. CircCRIM1 Destabilized HLA-F mRNA via Competing Interaction with IGF2BP1 in NSCLC

Next, we sought to investigate the molecular mechanism of regulating immune evasion underlying the role of circCRIM1 interaction with IGF2BP1 in NSCLC. RNA sequencing was performed, and the results revealed that 229 and 212 mRNAs were downregulated in overexpressed circCRIM1 A549 and Calu1 cells, respectively, compared with the vector. (Figure 4A,B). Among these differentially expressed mRNAs, only HLA-F was an immune-related gene and downregulated in both A549 and Calu1 cells (Figure 4B). Further qRT-PCR, Western blot, and flow cytometry assays confirmed that overexpressed circCRIM1 could reduce the expression levels of HLA-F (Figure 4C–E). Thus, the results of rescue experiments showed that IGF2BP1 significantly rescued the increased expression of HLA-F mediated by circCRIM1 in A549 and Calu1 cells, which demonstrated the interplay between circCRIM1 and IGF2BP1 in regulating the expression of HLA-F (Figure 4F,G and Appendix A).

In view of previous studies, which revealed that IGF2BP1 could regulate the stability of target mRNAs [17], we speculated that IGF2BP1 could stabilize HLA-F mRNA. To further probe the role of interaction between circCRIM1 and IGF2BP1 in altering the stability of HLA-F mRNA, the ActD assay was applied. The results showed that circCRIM1 overexpression markedly reduced the stability of HLA-F mRNA in A549 and Calu1 cells, and IGF2BP1 could rescue the decreased stability of HLA-F mRNA (Figure 5A,B and Appendix A). Moreover, the competing interaction with IGF2BP1 of circCRIM1 and HLA-F mRNA in NSCLC was confirmed by RIP assays and luciferase reporter assays. Moreover, the binding of HLA-F mRNA and IGF2BP1 was significantly reduced by ectopic expression of circCRIM1, and RRM domains could potentially contribute to the bonding (Figure 5C and Appendix A). To define the binding site of the interactions between HLA-F mRNA and IGF2BP1, we used the catRAPID website to predict and construct luciferase plasmids of wild-type and the binding site mutant fragment of HLA-F mRNA (Appendix A). Luciferase reporter assays showed that overexpressed circCRIM1 reduced the luciferase activity of reporters with wild-type HLA-F mRNA, but not with the binding site mutant HLA-F mRNA (Figure 5D). Furthermore, a decrease in luciferase activity mediated by circCRIM1 could be rescued by overexpressed IGF2BP1 (Figure 5E). In addition, rescue experiments verified that HLA-F overexpression significantly reversed the inhibition of immune evasion mediated by circCRIM1 in A549 cells (Appendix A). To summarize, these findings indicated that circCRIM1 destabilized HLA-F mRNA via competing interactions with IGF2BP1 to inhibit the immune evasion of NSCLC.

### 3.6. CircCRIM1 Is a Potential Therapeutic Target to Inhibit NSCLC Progression

To determine whether circCRIM1 could inhibit NSCLC progression by weakening the ability of immune evasion in vivo, hu-PBMC immune construction SCID mice were injected subcutaneously with A549 cells stably transfected with circCRIM1 or the control vector. Strikingly, we observed that the tumor volume markedly decreased in the circCRIM1 overexpressed group compared with the vector group (Figure 6A–C). Correspondingly, lower expression of HLA-F, more CD8+ T cell and NK cell infiltration, and more apoptosis tumor cells in tumors of the circCRIM1-overexpressing group were detected with the IHC assay (Figure 6D,E). Moreover, circCRIM1 overexpression increased the infiltration proportion of Granzyme B, IFN-γ, and TNF-α positive CD8+ T cells and NK cells in the subcutaneous tumor via flow cytometry (Figure 6F and Appendix A). Altogether, these data suggested that circCRIM1 is a potential immunotherapeutic target to inhibit NSCLC progression.

## 4. Discussion

Tumor cells may avoid being recognized and eliminated by the immune system through various mechanisms, including downregulating MHC molecules to achieve immune escape during the process of growth and metastasis [35,36]. It is significant to fully understand the molecular mechanism of tumor immune evasion for the development of more effective immunotherapeutic strategies [37]. Herein, we identified that circCRIM1 was a key regulator of immune evasion. Our study revealed that circCRIM1 was obviously downregulated in NSCLC tissues, and expression was positively correlated with favorable prognoses. Mechanistically, circCRIM1 bound to the IGF2BP1 protein and then destabilized the HLA-F mRNA, ultimately contributing to the attenuation of lung cancer immune evasion. These findings uncovered circCRIM1 as tumor suppressors via the regulation of tumor immunity and provided further evidence that circRNAs are extremely important in the immune evasion of NSCLC.

It is evident that circRNAs have been well-characterized in a variety of human diseases, including tumorigenesis, neurological diseases, cardiovascular diseases, and metabolic disorders [7,8,9]. Recently, circRNAs have been reported to play a potential part in regulating tumor immunity. For instance, exogenously purified circRNAs may act as antigens to activate innate immunity via the retinoic acid-inducible gene I (RIG-I) pathway in vitro [13]. Likewise, circNDUFB2 may elicit the activation of RIG-I-MAVS signaling cascades to induce antitumor immunity in NSCLC [18]. Furthermore, circRNAs may regulate the activity or cytotoxicity of immune cells in the TME [14,38,39]. For example, in hepatocellular carcinoma (HCC), circARSP91 upregulated the expression of UL16 binding protein 1 (ULBP1) to enhance the cytotoxicity of NK cells [38]. In addition, in pancreatic cancer (PC), the circ-0000977/miR-153/HIF1α axis promoted the immune evasion of PC cells by inhibiting the cytotoxicity of NK cells [39]. In the present study, we identified that circCRIM1 was mainly localized in the cytoplasm. Gain- and loss-of-function studies demonstrated that overexpressed circCRIM1 suppressed the immune evasion of NSCLC and promoted the expressions of Granzyme B, IFN-γ, and TNF-α of CD8+ T and NK cells. Although circRNAs related to tumor immunity regulation have been partially identified, circCRIM1 was distinguished by its role in influencing tumor immune evasion by combining with IGF2BP1. More importantly, our results proposed that circRNAs could modulate tumor immune evasion by interacting with important RNPs, providing a novel target for immunotherapy. Furthermore, circCRIM1 could act as a strong immune sensitizer, suggesting new targets for clinical treatments. In NSCLC, most studies focused on the role of circRNAs as miRNA sponges, but in our experiments, we found that circCRIM1 could bind to IGF2BP1 protein in the cytoplasm to exert its effect. However, other potential roles of circCRIM1 in the cytoplasm still need to be further clarified.

Previous studies have identified that IGF2BP1 plays an essential role in carcinogenesis in multiple solid tumors and that the high expression of IGF2BP1 is associated with poor prognoses [40,41,42]. Moreover, recent studies uncovered that circRNAs could interact with IGF2BP1 to promote the progression of NSCLC [18,43]. For instance, circXPO1 could enhance CTNNB1 mRNA stability by binding with IGF2BP1 to promote LUAD progression. However, little was known about the tumor immune evasion regulated by IGF2BP1 in NSCLC. In the current study, we found that IGF2BP1 facilitated the stabilization of HLA-F mRNA to suppress the cytotoxic function of CD8+T and NK cells, suggesting the immunomodulatory roles of IGF2BP1 in NSCLC. Furthermore, RNA pull-down and RIP analysis demonstrated that circCRIM1 interacted with the RNA-recognition motifs (RRMs) in the N-terminal part of IGF2BP1 to destabilize targets mRNAs through forming the circCRIM1/IGF2BP1 complex. The competitive suppression between circCRIM1 and HLA-F mRNA was possibly dependent on affinity differences in binding to IGF2BP1. Here, we revealed the novel role of the circRNA/IGF2BP1 complex to regulate tumor immune evasion, which would broaden the understanding of circRNAs in tumor immunity.

The MHC-Ib molecules (HLA-E, HLA-F, and HLA-G) were not ubiquitously expressed but were more specialized in distinct tissues and in function [24]. Recent studies reported that a variety of tumor cells highly expressed HLA-E, which allowed them to escape the immune recognition of NK and CD8+ T cells by interactions with inhibitory receptors of the CD94-NKG2 family [44]. HLA-G, primarily expressed on trophoblasts lining the placenta, presented endogenous peptides to contribute to the maternal tolerance of the fetus by activating inhibitory receptors of NK cell receptors (NKRs) and T cell receptors (TCRs) [45]. However, the role that HLA-F plays in immune regulation remains unclear. In our study, we confirmed that increased HLA-F promoted the immune evasion of NSCLC by inhibiting the cytotoxicity of CD8+ T and NK cells. Furthermore, the expression of HLA-F could be regulated by the circCRIM1/IGF2BP1 complex. These findings facilitate our understanding of HLA-F-regulated immune evasion in NSCLC.

In summary, our data demonstrate that circCRIM1 is an important tumor suppressor for NSCLC. These findings indicate that circCRIM1 plays a critical role in the immune evasion of NSCLC. Therefore, the mechanistic characterization of circCRIM1 and its functional crosstalk with IGF2BP1 may help to pave the way to developing a novel immunotherapy target for NSCLC.

## 5. Conclusions

This study identifies circCRIM1 as a new tumor suppressor that inhibits tumor immune evasion through a competitive combination with IGF2BP1 to destabilize HLA-F mRNA. This indicates that circCRIM1 may serve as an exploitable immunotherapy target for patients with NSCLC.

## Figures and Tables

**Figure 1 cells-12-00273-f001:**
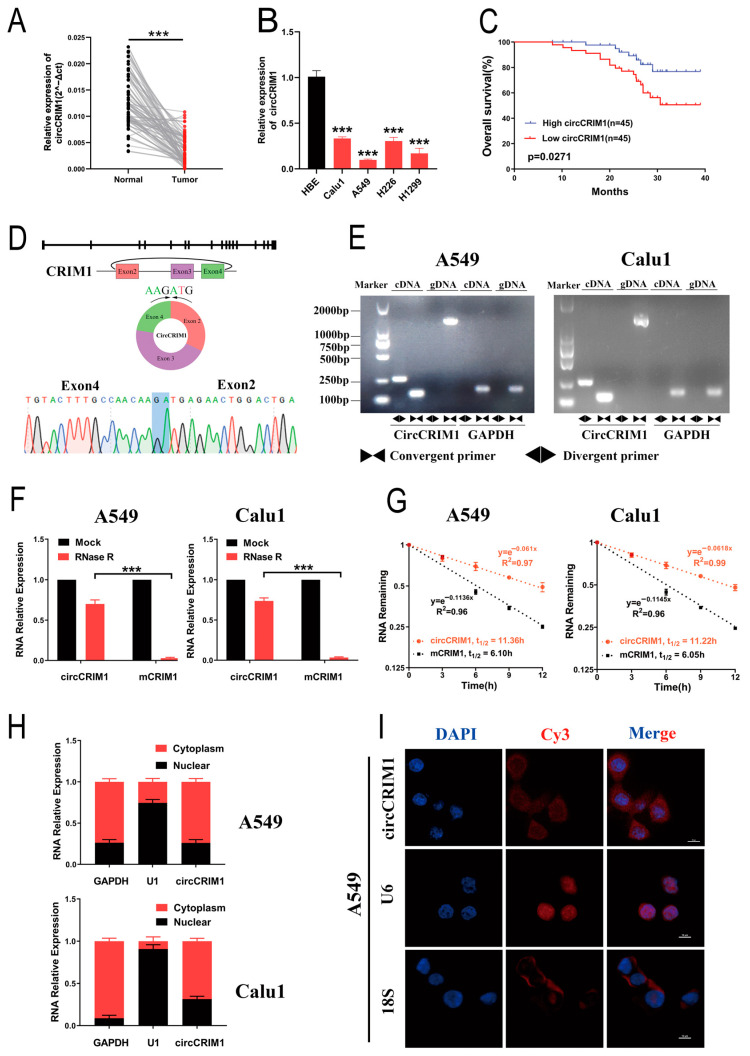
Identification and distribution of circCRIM1. (**A**) The expression of circCRIM1 was detected with qRT-PCR in 90 pairs of NSCLC and adjacent normal lung tissues, and GAPDH was used as an internal control. Statistical analysis by paired *t*-test. (**B**) The expression of circCRIM1 was detected with qRT-PCR in HBE, Calu1, A549, H226, and H1299 cells. Statistics by one-way ANOVA. (**C**) Kaplan–Meier curves of OS in NSCLC patients. Patients were grouped by the median circCRIM1 expression. The *p* value was calculated using a log-rank test. (**D**) Scheme illustrating the production and sequencing analysis of head-to-tail splicing junction in circCRIM1. (**E**) The existence of circCRIM1 was validated in A549 and Calu1 lung cancer cell lines via qRT-PCR. Divergent primers amplified circCRIM1 in cDNA but not genomic DNA (gDNA). GAPDH was used as a negative control. (**F**) Relative RNA levels were analyzed via qRT-PCR in A549 and Calu1 cells treated with or without RNase R (n = 3). Statistics analysis by unpaired *t*-test. (**G**) The relative RNA levels of circCRIM1 and mCRIM1 were analyzed using qRT-PCR after treatment with actinomycin D at the indicated time points in A549 and Calu1 cells (n = 3). (**H**) Identification of circCRIM1 cytoplasmic and nuclear distribution was carried out using qRT-PCR analysis in A549 and Calu1 cells. GAPDH and U1 were applied as positive controls in the cytoplasm and nucleus, respectively (n = 3). (**I**) Identification of circCRIM1 cytoplasmic and nuclear distribution via FISH in A549 cells. 18S and U6 were applied as positive controls in the cytoplasm and nucleus, respectively; circCRIM1, 18S, and U6 probes were labeled Cy3; nuclei were stained with DAPI. Data shown as mean ± SD; ns, not significant; *** *p* < 0.001.

**Figure 2 cells-12-00273-f002:**
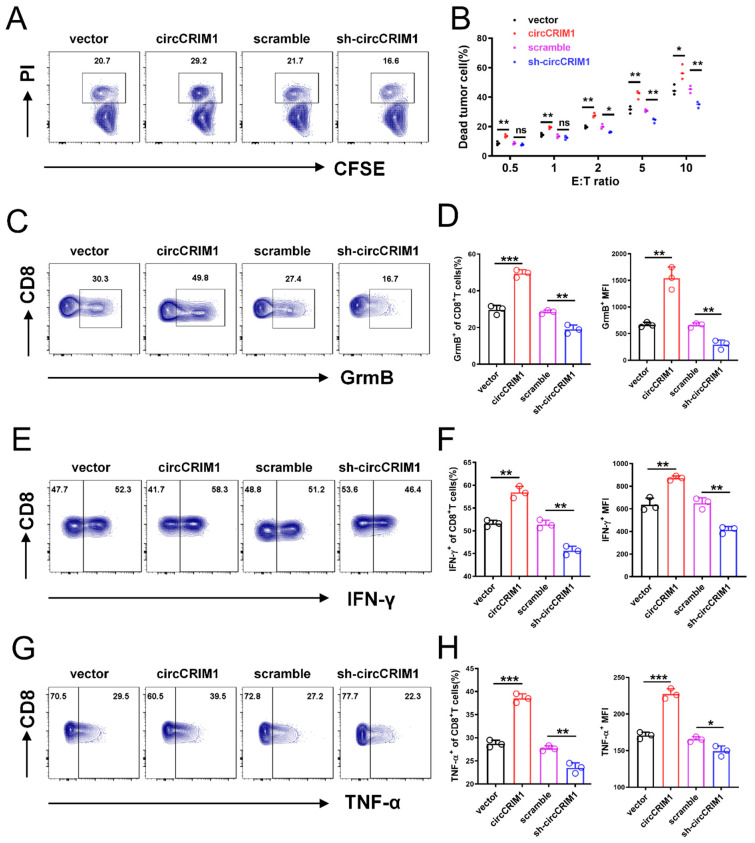
CircCRIM1 inhibits the immune evasion of lung cancer cells from the killing of CD8+ T cells. (**A**) A549 cells that were stably transfected with vector, sh-circCRIM1#2, and circCRIM1, and co-cultured with activated CD8+T cell for 24 h. E:T ratio (effector to target cell ratio) = 2:1. (**B**) Flow cytometry analysis displaying the percentage of dead tumor cells in co-culture assays with different E:T ratios. Statistical analysis by unpaired *t*-test. (**C**–**H**) Flow cytometry analysis of Granzyme B, IFN-γ, and TNF-α expression of CD8+ T cells co-cultured with A549 cells for 24 h. E:T ratio = 2:1. Statistical analysis by one-way ANOVA. Data shown as mean ± SD; ns, not significant; * *p* < 0.05, ** *p* < 0.01, *** *p* < 0.001.

**Figure 3 cells-12-00273-f003:**
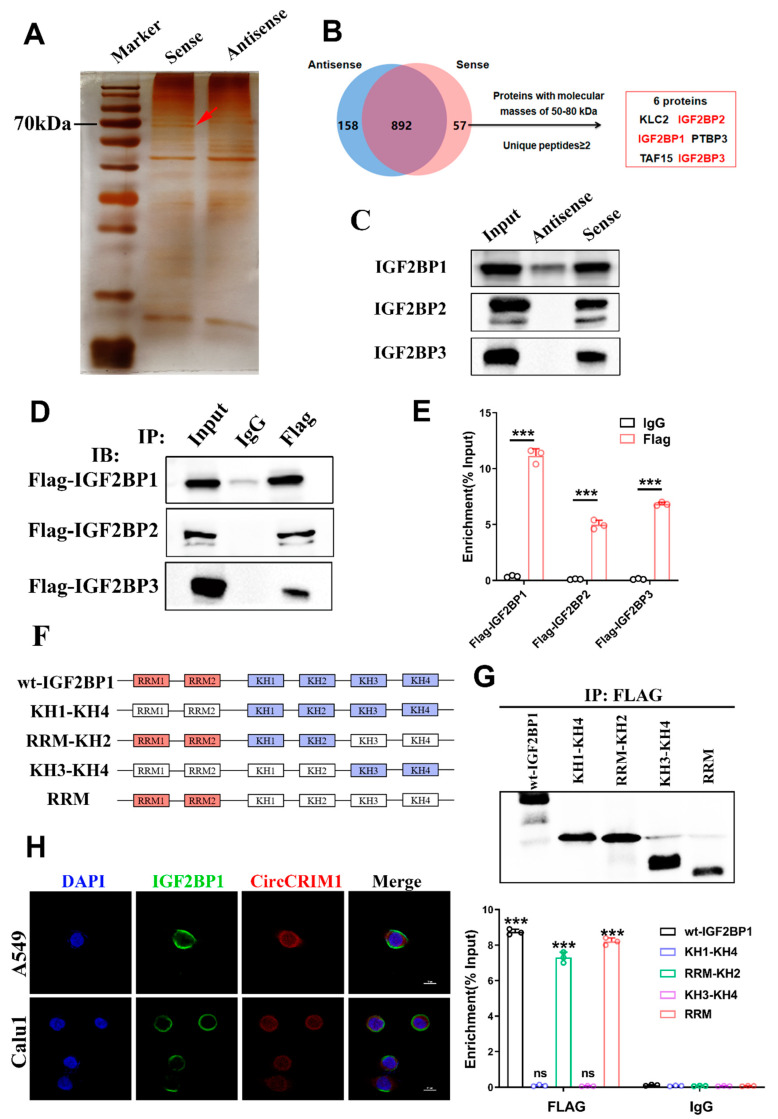
CircCRIM1 binds to IGF2BP1 protein. (**A**) Biotin-labeled sense or antisense circCRIM1 probes were used for RNA-protein pull-down against A549 cell lysates. Identification of proteins that interact with circCRIM1 via silver staining. Red arrows indicate the major differential band precipitated in A549 lysates. (**B**) Analysis pipeline was performed to identify proteins that interact with circCRIM1 according to mass spectrometry assay. (**C**) IGF2BP1/2/3 immunoblot analysis of the biotin-labeled sense and antisense circCRIM1 probes pull-down eluate from lysates of A549 cells. (**D**) RNA immunoprecipitation (RIP) assays in A549 cells using IGF2BP1/2/3 and IgG antibody. (**E**) The FLAG-IGF2BP1/2/3-enriched circCRIM1 relative to the IgG-enriched value was calculated via qRT-PCR. Statistics by unpaired *t*-test. (**F**) Schematic diagram of full-length or truncations FLAG-tagged of IGF2BP1protein. Wt-IGF2BP1 (full-length IGF2BP1), KH1-KH4 (from KH1 to KH4 domain), RRM-KH2 (from RRM1 to KH2 domain), KH3-KH4 (from KH3 to KH4 domain), RRM (from RRM1 to RRM2 domain). (**G**) RIP analysis for circCRIM1 enrichment in A549 cells transiently transfected with plasmids containing the indicated FLAG-tagged full-length or truncated constructs. Relative enrichment of endogenous circCRIM1 in FLAG-tagged full-length or truncated IGF2BP1 RIP was measured using qRT-PCR. Statistics by unpaired *t*-test. (**H**) Dual RNA-FISH and immunofluorescence staining assays indicating the co-localization of circCRIM1 (red) and IGF2BP1 (green), with nuclei staining with DAPI (blue). Data shown as mean ± SD; ns, not significant; *** *p* < 0.001.

**Figure 4 cells-12-00273-f004:**
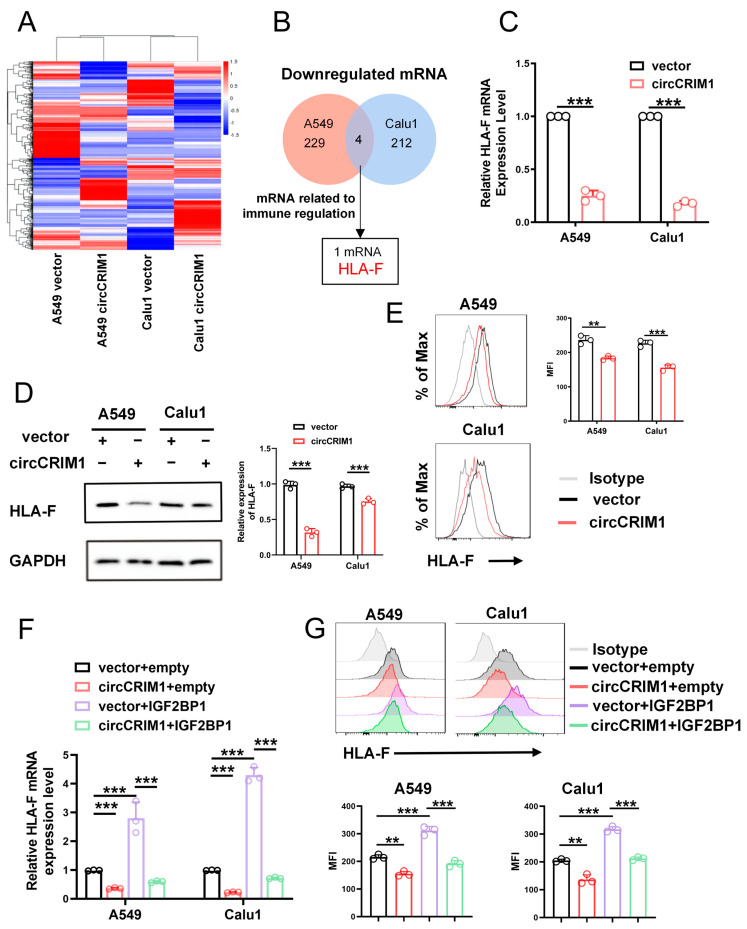
CircCRIM1 downregulates HLA-F expression via interactions with IGF2BP1 in NSCLC cells. (**A**) Heat map showing the differentially expressed genes upon circCRIM1 overexpression in A549 and Calu1 cells. Each sample contained a mixture of three repeats. (**B**) Venn diagram indicating the discovery of immune-related mRNA both downregulated in A549 and Calu1 cells. (**C**–**E**) qRT-PCR analysis (**C**), Western blot analysis (**D**) and flow cytometry analysis (**E**) comparing circCRIM1-overexpressed A549 and Calu1 cells with their respective control cells were shown for relative expression of HLA-F. GAPDH was used as a loading control. Statistics by unpaired *t*-test (**C**,**E**). (**F**,**G**) qRT-PCR analysis and flow cytometry analysis showing the expression of HLA-F in A549 and Calu1 cells stably transfected with vector or circCRIM1, and co-transfected with IGF2BP1 or empty vector. Statistics by one-way ANOVA with Tukey’s multiple comparison analysis. Data shown as mean ± SD; ns, not significant; ** *p* < 0.01, *** *p* < 0.001.

**Figure 5 cells-12-00273-f005:**
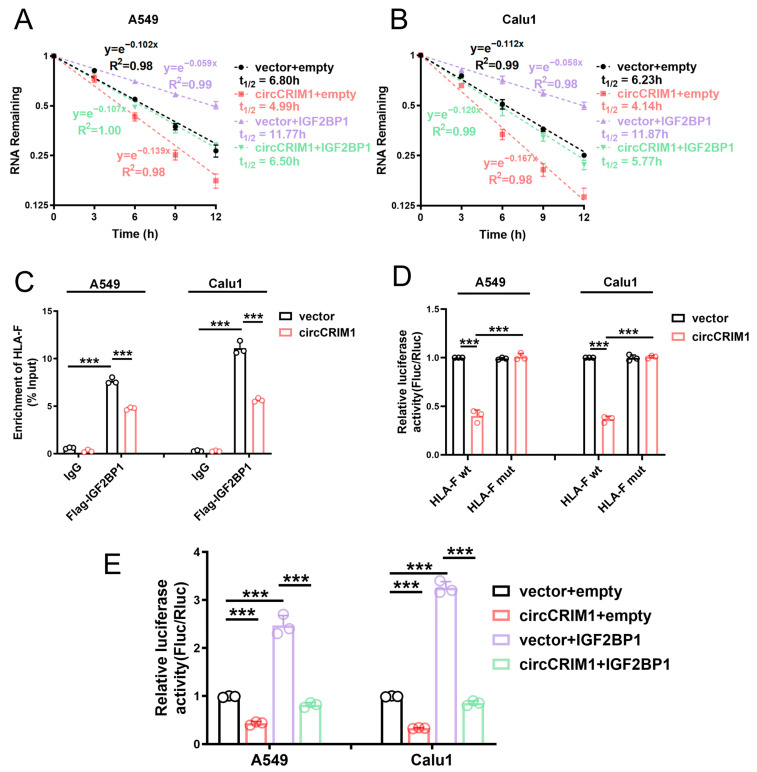
CircCRIM1 affects the mRNA stability of HLA-F via competing interactions with IGF2BP1 in NSCLC. (**A**,**B**) The relative RNA levels of HLA-F mRNA were analyzed via qRT-PCR and the mRNA half-life was calculated after treatment with actinomycin D at the indicated time points in A549 and Calu1 cells stably transfected with vector or circCRIM1, and co-transfected with IGF2BP1 or empty vector. Data expressed as mean ± SD of three independent experiments. (**C**) RIP-qPCR showed endogenous IGF2BP1 or recombinant IGF2BP1 binding to HLA-F transcripts in A549 and Calu1 cells stably transfected with vector or circCRIM1. Statistical analysis by one-way ANOVA. (**D**) Relative luciferase activity of wild-type (HLA-F wt) or binding site mutant (HLA-F mut) HLA-F mRNA reporters in A549 cells, with or without ectopic expression of circCRIM1. Statistical analysis by one-way ANOVA. (**E**) Relative luciferase activity of HLA-F mRNA in A549 and Calu1 cells stably transfected with vector or circCRIM1, and co-transfected with IGF2BP1 or empty vector. Statistics by one-way ANOVA with Tukey’s multiple comparison analysis. Data shown as mean ± SD; ns, not significant; *** *p* < 0.001.

**Figure 6 cells-12-00273-f006:**
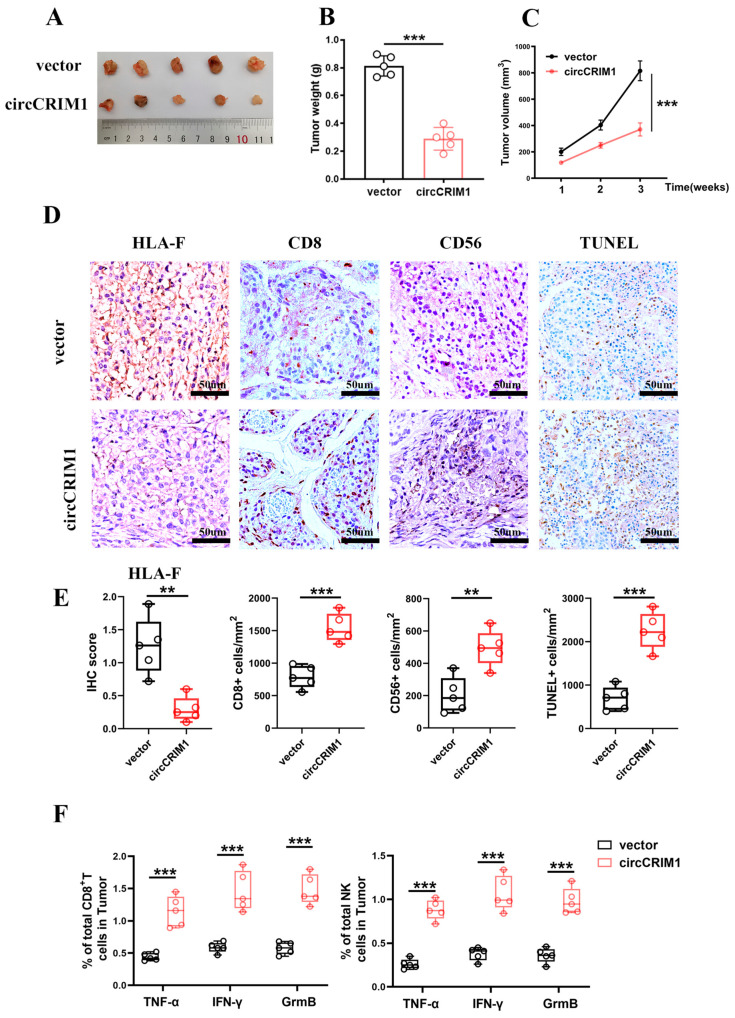
Biological implications of circCRIM1 in NSCLC. (**A**–**C**) The volume and weight of subcutaneous xenograft tumors (n = 5 mice per group). Statistical analysis by unpaired *t*-test (**B**) and two-way ANOVA (**C**). (**D**,**E**) Representative images of immunohistochemical images of HLA-F, CD8, and CD56, and TUNEL staining on the tumor tissues from xenografts mice subcutaneously injected with A549 cells stably transfected with vector or circCRIM1. Boxplots showing the IHC scores for HLA-F and the percentage of positive cells for CD8, CD56, and TUNEL. Statistical analysis by unpaired *t*-test (**E**). (**F**) Boxplots of representative flow cytometric analysis displaying the absolute percentages of Granzyme B, IFN-γ and TNF-α positive CD8+ T cells and NK cells in the tumor. Statistical analysis by unpaired *t*-test. Data shown as mean ± SD; ** *p* < 0.01, *** *p* < 0.001.

## Data Availability

The datasets used or analyzed during the current study are available from the corresponding author upon reasonable request.

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
