# Peer review of "Silencing of circCRIM1 Drives IGF2BP1-Mediated NSCLC Immune Evasion"

_cells, 2023, doi:10.3390/cells12020273_

Round 1

Reviewer 1 Report

File uploaded

Reviewer 2 Report

The manuscript “Silencing of circCRIM1 drives IGF2BP1-mediated NSCLC immune evasion” by Peng et al., investigated how circaCRIM1 regulated target mRNA especially HLA-F mRNA through binding competition with IGF2BP1 in lung cancer model system. The authors performed several essential experiments to validate the existence of circCRIM1 expression and localization in two cell lines and also tested circCRIM1 functions in the immune evasion of lung cancer cells. They found circCRIM1 binding proteins by RNA-protein pull-down assay I think this are most significant findings in this study. Then authors investigated to find the target HLA-L mRNA that most important question in this field. Overall experimental design and results were extremely clear. I don’t have any further comments. Thank you.  

Round 2

Reviewer 1 Report

The authors responded to all the major and minor points in a short period. They provide relevant and precise modifications that strongly strenghten teh manuscript.